# Ghrelin Alleviates Experimental Ulcerative Colitis in Old Mice and Modulates Colonocyte Metabolism via PPARγ Pathway

**DOI:** 10.3390/ijms24010565

**Published:** 2022-12-29

**Authors:** Srilakshmi Muthyala, Robert S. Chapkin, Chaodong Wu, Chia-Shan Wu

**Affiliations:** Department of Nutrition, Texas A&M University, College Station, TX 77843, USA

**Keywords:** inflammatory bowel disease, gut inflammation, colonocyte metabolism, ghrelin, PPARgamma

## Abstract

There is a growing prevalence of inflammatory bowel disease (IBD), a chronic inflammatory condition of the gastrointestinal tract, among the aging population. Ghrelin is a gut hormone that, in addition to controlling feeding and energy metabolism, has been shown to exert anti-inflammatory effects; however, the effect of ghrelin in protecting against colitis in old mice has not been assessed. Here, we subjected old female C57BL/6J mice to dextran sulfate sodium (DSS) in drinking water for six days, then switched back to normal drinking water, administered acyl-ghrelin or vehicle control from day 3 to 13, and monitored disease activities throughout the disease course. Our results showed that treatment of old mice with acyl-ghrelin attenuated DSS-induced colitis. Compared to the DSS group, ghrelin treatment decreased levels of the inflammation marker S100A9 in the colons collected on day 14 but not on day 8, suggesting that the anti-inflammatory effect was more prominent in the recovery phase. Ghrelin treatment also significantly reduced F4/80 and interleukin-17A on day 14. Moreover, acyl-ghrelin increased mitochondrial respiration and activated transcriptional activity of the peroxisome proliferator-activated receptor gamma (PPARγ) in Caco-2 cells. Together, our data show that ghrelin alleviated DSS-induced colitis, suggesting that ghrelin may promote tissue repair in part through regulating epithelial metabolism via PPARγ mediated signaling.

## 1. Introduction

The prevalence of inflammatory bowel disease (IBD), a chronic inflammatory condition of the gastrointestinal tract, continues to rise among the general United States population and worldwide [1]. IBD includes Crohn’s disease and ulcerative colitis. Both conditions are chronic, relapsing, and remitting; the precise etiology remains unclear, and a cure for IBD has yet to be discovered. Furthermore, there is a growing prevalence of IBD among the older population (>65 years of age) [2,3]. Unfortunately, the pathogenesis of older-onset IBD is not well defined [2,4].

The colonic tissue microenvironment helps to maintain a symbiotic microbial community. Increasing evidence suggests a critical role of gut microbiota dysbiosis in contributing to inflammation and dysregulated immune systems in IBD [5,6]. The IBD gut is characterized by reduced microbiome diversity and depletion of protective bacteria [7,8], coinciding with an expansion of pro-inflammatory microbes such as Enterobacteriaceae [9,10]. Furthermore, altered colonocyte metabolism and increased oxygen levels in the IBD gut facilitate the expansion of facultative microbes [11,12,13]. Interestingly, recent findings suggest the peroxisome proliferator-activated receptor gamma (PPARγ) as a critical molecular determinant modulating host-microbe interactions during IBD by regulating colonic metabolism to maintain low epithelial oxygenation, thus preventing the expansion of dysbiotic microbes [13,14]. Therefore, factors regulating PPARγ activity and/or colonocyte metabolism could potentially provide therapeutic benefits in IBD.

Ghrelin, a 28 amino acid peptide hormone produced mainly by the X/A-like cells in the gastrointestinal tracts [15,16], plays multi-faceted roles in nutritional balance, metabolism, and inflammation [17,18]. We previously reported that old ghrelin knockout mice showed significantly increased intestinal permeability and inflammation, reduced microbially-derived indole metabolites, and developed exacerbated disease activities when subjected to dextran sulfate sodium (DSS)-induced experimental colitis [19]. Furthermore, previous studies have demonstrated the anti-inflammatory effects of ghrelin in multiple pathological conditions, including endotoxic shock [20], sepsis [21], and experimental colitis [22,23,24,25]; however, the cellular and molecular mechanisms mediating the therapeutic effects of ghrelin are less clear.

We previously showed that 500 μg/kg acyl-ghrelin (AG) treatment alleviated fasting-induced muscle atrophy in old male mice [26]. Here we used the same dose of AG to test its efficacy in attenuating DSS-induced colitis in old female mice. Of note, given the importance of the gut microbiome in contributing to the pathogenesis of ulcerative colitis, experimental mice were maintained on semi-purified diets to ensure consistency and reproducibility of the disease course. To explore the potential effect of ghrelin on colonocyte metabolism, we assessed the oxygen consumption rate in a human colon adenocarcinoma cell line (Caco-2 cells). Furthermore, we tested whether PPARγ is a downstream factor mediating the effect of ghrelin using a PPARγ reporter assay in Caco-2 cells. Our data suggest a novel role for ghrelin in regulating intestinal epithelial metabolism through PPARγ.

## 2. Results

### 2.1. Ghrelin Treatment Attenuated Some Clinical Symptoms of DSS-Induced Colitis in Old Mice

We subjected 16–18-months-old female C57BL/6J mice to DSS in drinking water for 6 days (days 0–6), then switched back to normal drinking water, administered AG or vehicle control from day 3 to 13, and monitored disease activities throughout the disease course to assess the effects of ghrelin on disease progression and resolution in old mice (Figure 1A). In terms of body weight, both DSS and DSS + AG groups showed a similar degree of body weight loss during disease progression, while DSS + AG group showed a trend for increased body weight recovery from day 9 to day 14 (Figure 1B). In terms of fecal consistency score, the DSS + AG group showed significantly improved fecal consistency in the late recovery phase (days 12–14), while the DSS-treated old female mice still showed softened feces (Figure 1C). Interestingly, ghrelin treatment significantly reduced rectal bleeding during the peak of gut inflammation (days 6–8) compared to the DSS group (Figure 1D). Assessment of colon length showed that DSS treatment significantly reduced colon length on day 8 and day 14 compared to that of the control, with or without ghrelin treatment (Figure 1E). On the other hand, DSS-treated mice showed significantly increased colon weight on day 8, possibly due to edema, and ghrelin treatment significantly reduced colon weight (Figure 1F). By day 14, the weights of the colons from DSS-treated mice had recovered to that comparable to the control (Figure 1F).

### 2.2. Ghrelin Treatment Reduced Colon Inflammation

Next, we assessed the tissue morphology of colon Swiss rolls collected on day 14 to evaluate the extent of tissue recovery. Hematoxylin and eosin (H&E) staining showed significant edema in the submucosa and hyperplasia of epithelial cells in the colons of DSS-treated mice compared to that of control mice, suggesting lingering gut inflammation in old mice after an episode of colitis (representative images shown in Figure 2A). These signs of gut inflammation were less apparent in the DSS + AG group, with improved overall tissue morphology, suggesting that ghrelin treatment reduced colon inflammation and promoted tissue repair (Figure 2A). Immunoblot analysis of colons collected on day 14 showed that levels of S100A9 protein, a subunit of the gut inflammation biomarker Calprotectin, were significantly increased in the colons of DSS-treated mice, and there was a trend for reduction by ghrelin treatment (Figure 2B,C). On the other hand, colons collected on day 8 showed elevated levels of S100A9 in both DSS and DSS + AG groups, suggesting similar inflammatory levels in the acute phase (Appendix A). Importantly, levels of F4/80 protein, a biomarker of macrophages, were significantly increased in the colons of DSS-treated mice, and ghrelin treatment significantly reduced F4/80 proteins, suggesting decreased amounts of macrophages (Figure 2B,D).

Next, we assessed inflammatory cytokines and chemokines in colon tissues collected on day 14. Colons of DSS-treated mice showed significantly increased levels of interleukin (IL)-17A, which was significantly reduced by ghrelin treatment (Figure 3A). On the other hand, there were no significant differences in the T-helper 1 (Th1) cytokines tumor necrosis factor (TNF)α or the Th2-cytokine IL-10 among the groups on day 14 (Figure 3B,C). The Th2-cytokine IL-4 was significantly reduced in the colons of DSS-treated mice, while there was a trend for higher IL-4 levels in the colons of the DSS + AG group (Figure 3D). The chemokine keratinocyte chemoattractant (KC) remained significantly elevated in DSS and DSS + AG groups on day 14 (Figure 3E). Interestingly, IL-2 levels were significantly reduced in the colons of DSS-treated mice, while ghrelin treatment significantly restored the levels, albeit not completely to that observed in the control mice (Figure 3F). Taken together, these data show that DSS-treated old female mice exhibited elevated F4/80 protein expression and IL-17A in the colon tissue microenvironment on day 14, and ghrelin treatment significantly reduced these markers of inflammation.

### 2.3. Ghrelin Increased Colonocyte Mitochondrial Function and PPARγ Transcriptional Activity

Previously we showed that ghrelin-deficient mice exhibited impaired mitochondrial function in muscle, and at the cellular level, ghrelin increased oxygen consumption rate (OCR) in C2C12 muscle cells [26]. To determine whether ghrelin also exerts a cell-autonomous effect on intestinal epithelial cells, we tested the effect of ghrelin on bioenergetics in human Caco-2 cells using extracellular metabolic flux assays. Indeed, 24 h of treatment with 100 nM human AG increased baseline mitochondrial OCR as well as maximal OCR compared to vehicle control in Caco-2 cells (Figure 4A).

A recent study reported that 5-aminosalicylic acid (5-ASA), a PPARγ agonist widely used as a first-line medication for the treatment of ulcerative colitis [27], has been shown to increase mitochondrial respiratory capacity in Caco-2 cells, which was indicative of increased mitochondrial bioenergetics [14]. To test whether PPARγ is a potential mediator of ghrelin’s effect on mitochondrial function in Caco-2 cells, we utilized a PPARγ-reporter assay (Figure 4B). Ghrelin dose-dependently activated PPARγ transcriptional activity (Figure 4B). Furthermore, the increase in transcriptional activation was abolished by co-administration of a growth hormone secretagogue receptor (GHSR) antagonist JMV-2959, suggesting that ghrelin signals through its receptor GHSR to activate PPARγ transcriptional activity in Caco-2 cells. When taken together, these data suggest that ghrelin can alter colonic metabolism, likely via activation of PPARγ in intestinal epithelial cells lining the colon.

## 3. Discussion

This study aimed to evaluate the potential therapeutic effect of ghrelin in DSS-induced colitis in old mice. Our results show that treatment of old mice with AG attenuated DSS-induced colitis, and decreased gut inflammation on day 14, eight days after cessation of DSS in drinking water. Interestingly, at this time point, even though clinical symptoms of disease activities such as diarrhea and rectal bleeding had receded in old DSS-treated mice, there were still marked signs of colon inflammation, indicated by elevated levels of S100A9, F4/80 and pro-inflammatory cytokine IL-17A. This contrasted with a previous study employing young mice in a DSS-induced colitis model where disease symptoms and gut inflammation resolved at this time point [28]. In their study, Czarnewski et al. [28] performed an unbiased characterization of the inflammatory and tissue repair processes in young (8–12-weeks–old) female C57BL/6J mice following administration of 2.5% DSS in drinking water for 7 days and letting mice recover for 7 days. They assessed the transcriptomic profile of colons collected every second day in the entire disease course and applied the identified gene networks to perform unsupervised and biologically driven analysis of highly variable human data sets. Their study demonstrated that the DSS-induced colitis model is a relevant model for studying human ulcerative colitis. Importantly, their time-series transcriptional characterization of colitis identified epithelial cell function and PPARγ signaling among the key pathways downregulated during the acute (days 6–10) and recovery phases (days 12–14) of DSS-induced inflammation. Taken together, these data suggest that age is an important factor in determining the time course of disease resolution, and ghrelin treatment attenuates inflammation in the recovery phase of DSS-induced colitis in old mice.

Our data show that ghrelin reduces inflammation markers at day 14 but not on day 8, while rectal bleeding of the DSS + AG group exhibit a significant decrease on day 6 and day 8. While DSS is widely used in mouse model of colitis, detailed mechanisms by which DSS induces intestinal inflammation and accompanying diarrhea and bleeding are unclear. It has been suggested that the negatively-charged DSS forms nano-lipocomplexes with medium-chain-length fatty acids in the colon and induces erosion of the epithelia that ultimately compromises barrier integrity, resulting in increased colonic epithelial permeability and translocation of pro-inflammatory intestinal contents (bacteria and their products) into underlying tissue [29,30]. In addition, the anticoagulant property of DSS aggravates intestinal bleeding. Hence, the cessation of DSS from day 6 allows tissue restitution without further aggravation, and this process may precede the clearance of infiltrating immune cells and the resolution of inflammation. The observation that ghrelin reduced rectal bleeding on days 6 and 8 suggests that ghrelin may also mediate this early tissue restitution phase via regulation of proliferation and/or apoptosis. Similarly, Zhang et al. [25] recently reported that ghrelin inhibits intestinal epithelial cell apoptosis in DSS-induced colitis. Interestingly, a recent study demonstrated a novel mechanism involving neutrophil-mediated immunothrombosis and extracellular trap formation to resolve rectal bleeding in ulcerative colitis [31]; whether ghrelin modulates neutrophils in the process of immunothrombosis to ameliorate rectal bleeding remains to be determined.

Our data demonstrate, for the first time, that ghrelin may be a novel regulator of intestinal epithelial metabolism via the regulation of PPARγ activity. Ghrelin dose-dependently activated PPARγ transcriptional activity, and increased oxygen consumption rate in Caco-2 cells. PPARγ is a member of the nuclear receptor superfamily of ligand-activated transcription factors [32,33] and is highly expressed in adipose tissue and colonic epithelium, and to a lesser extent in muscle, macrophage, T and B cells [34,35]. Ligands for PPARγ include natural compounds with relatively low affinity, such as polyunsaturated fatty acids, oxidized low-density lipoprotein, certain eicosanoids, as well as drugs such as the thiazolidinedione (TZD), a class of insulin-sensitizing drugs used as second-line oral therapy for diabetes [36,37,38,39]. PPARγ is a critical regulator of adipocyte biology and lipid and carbohydrate metabolism [40,41,42]. In preclinical models of IBD, activation of PPARγ by conjugated linoleic acid suppresses gut inflammatory lesions, weight loss and inflammatory mediator expression [43]. Clinically, the PPARγ agonist rosiglitazone showed therapeutic efficacy in patients with ulcerative colitis [44]. Mice lacking PPARγ in the colonic epithelium displayed increased susceptibility to DSS-induced experimental IBD, histological lesions and elevated levels of the pro-inflammatory cytokines IL-6, IL-1β, TNFα [45,46]. More recently, Byndloss et al. demonstrated that PPARγ signaling in the intestinal epithelial cells drives the energy metabolism of these cells toward β-oxidation and suppresses the synthesis of inducible nitric oxide synthase (iNOS) and oxygen bioavailability. As a result of this PPARγ signaling, the anaerobic milieu in the colon is maintained, which prevents the growth of facultative, pro-inflammatory microbes [13,47]. Furthermore, 5-ASA, a PPARγ agonist and widely used as a first-line medication for the treatment of ulcerative colitis [27], has been shown to increase the spare mitochondrial respiratory capacity in Caco-2 cells, indicative of increased mitochondrial bioenergetics [14]. Taken together, these data suggest that PPARγ signaling regulates colonic metabolism to maintain low epithelial oxygenation, preventing the expansion of dysbiotic microbes. This is noteworthy because our in vitro results suggest that PPARγ may be a downstream factor mediating the effect of ghrelin on colonocyte metabolism. Future work testing whether targeted disruption of intestinal epithelial cell PPARγ abolishes ghrelin’s therapeutic effect in attenuating DSS-induced colitis in vivo is required to confirm this observation. In addition, complementary in vivo experiments employing hydroxy probes to assess epithelial oxygenation levels, isolation of intestinal epithelial cells from the mouse model to assess mitochondrial function or gene expression changes in β-oxidation pathways, and characterization of gut microbiota are needed to further elucidate the mechanism by which ghrelin ameliorates colitis.

A previous study has shown the therapeutic action of ghrelin in both chemical (2,4,6-trinitrobenzene sulfonic acid, TNBS) and DSS-induced colitis, experimental models of Crohn’s disease and ulcerative colitis, respectively [22]. The authors showed that ghrelin down-regulated Th1 cytokine response and stimulated IL-10 production in TNBS-induced colitis [22]. In addition to IL-10, ghrelin increased the production of TGF-β1 by activated lamina propria mononuclear cells and mesenteric lymph node cells; these data suggest that ghrelin favors the generation/activation of IL-10/TGF-β-secreting CD4+ Foxp3+regulatory T (Treg) cells [22]. Interestingly, while we observed no significant changes in IL-10 levels, the levels of IL-4, another Th2 cytokine, were restored in ghrelin-treated mice. Furthermore, IL-17A levels were significantly reduced by ghrelin treatment. Meta-analysis showed that genetic polymorphism and increased circulating levels of IL-17 contribute to the development and progression of ulcerative colitis [48]. We also observed a significant reduction in IL-2 levels in DSS-treated mice, which was partially reverted by ghrelin treatment. Previous studies have shown that low-dose IL-2 treatment improves the survival and expansion of peripheral blood Treg cells [49,50]. Consistently, low-dose IL-2 treatment alleviates DSS-induced colitis largely by inhibiting apoptosis and recovering intestinal integrity [51]. In line with these data, a recent study reported that ghrelin inhibited TNFα-induced apoptosis in Caco-2 cells and protected against DSS-induced colitis [25]. Taken together, these data suggest that ghrelin can exert both direct and indirect effects on different cell types; how the different cell types work in concert to reduce inflammation and promote tissue repair requires further mechanistic studies. Furthermore, a limitation of the current study is the use of a single dose of AG to test its efficacy in old mice. Future studies employing a more comprehensive dose range will be carried out to assess ghrelin’s therapeutic effects and validate the potential mechanisms in vivo.

In summary, our data suggest that ghrelin may be a novel regulator of intestinal epithelial metabolism via the PPARγ signaling pathway. We previously showed that chronic ghrelin deficiency is associated with exacerbated loss of gut barrier function, increased intestinal permeability, and reduced levels of microbially-derived indoles and related metabolites, suggesting gut dysbiosis [19]. Furthermore, both ghrelin and ghrelin-receptor-deficient mice showed increased susceptibility to DSS-induced colitis [19,52]. Together with the current study demonstrating the therapeutic effects of ghrelin treatment, the loss- and gain-of-function data strongly support the role of ghrelin in maintaining intestinal homeostasis.

## 4. Materials and Methods

### 4.1. Animals

C57BL/6J mice were raised in-house and maintained in the animal facility of Texas A&M University under controlled photoperiods (12:12 light-dark cycle, lights on at 6:00 AM) at 22 ± 2 °C. Mice were given ad libitum access to water and a regular chow diet (TD. 2018, Envigo, Madison, WI, USA, with 18%, 58% and 24% calories from fat, carbohydrates and protein, respectively). All experimental procedures were approved by the Institutional Animal Care and Use Committee at Texas A&M University (Protocol code: AUP 2019-0273), and all methods were performed in accordance with the relevant guideline and regulations.

### 4.2. Experimental Design

Female C57BL/6J mice from 16–18 months (M) of age were subjected to DSS-induced colitis (n = 5, 7 and 9 for control, DSS and DSS+acyl-ghrelin (AG) groups, respectively). To ensure a consistent microenvironment and minimize cage-to-cage variation, mice were placed in disposable micro-insulator cages two weeks prior to the start of the experiment, and two scoops of soiled beddings from all experimental cages were mixed, and one scoop of the mixed beddings was placed into the new cages containing fresh, autoclaved beddings. At the same time, mice were switched to a semi-purified OpenSource diet D12450J (Research Diets Inc., New Brunswick, NJ, USA) two weeks prior to the start of the experiment and maintained on this diet until termination; this diet contains 50 mg cellulose as a source of fiber. After two weeks of acclimatization, mice received normal drinking water (control group) or 2% DSS (36–50 kDa; MP Biomedicals, Irvine, CA, USA) in drinking water ad libitum for 6 days, then switched back to normal drinking water for 8 days; water was refreshed every 2–3 days (Figure 1A). Mice were intraperitoneally injected daily with 500 μg/kg AG (031–31; Phoenix Pharmaceuticals, Burlingame, CA, USA) or with vehicle saline solution in the mornings between 9–10:00 AM, after disease activity assessments. All mice were assessed for body weight, fecal consistency, and macroscopic fecal blood scores as described [19]. Animals were terminated on study day 8 (n = 4, 4 for DSS and DSS + AG groups, respectively) and day 14 (n = 5, 3 and 5 for control, DSS and DSS + AG groups, respectively) after final scoring. Colons were dissected, flushed with PBS, opened longitudinally, weighed and the length measured. The colon strip was cut length-wise, 1/3 was flash frozen for protein analysis, and 2/3 was Swiss-rolled, fixed in 4% paraformaldehyde, and processed for histology. The fixed Swiss-rolled colon tissues were routinely processed, embedded in paraffin, and cut into 5-μm sections. The sections were stained with H&E.

### 4.3. Cytokine Analyses

Snap frozen colon strips were weighed and homogenized at a 10:1 ratio (10 mL per 1 g of tissue) in tissue protein extraction reagent (T-PER, ThermoFisher Scientific, Waltham, MA, USA) with a complete protease inhibitor cocktail (Roche, Mannheim, Germany). Homogenates were centrifuged at 10,000 g for 5 min, and supernatants were stored at −20 °C until analysis. Protein concentrations were measured using the bicinchoninic acid (BCA) protein assay (ThermoFisher Scientific, Waltham, MA, USA) in duplicates. 25 μL of each sample was used for cytokine analyses using the Mouse Cytokine/Chemokine Magnetic Bead Panel Milliplex Map Kit (Millipore, Burlington, MA, USA), according to the manufacturer’s instructions. The plate was analyzed on a BioPlex 200 (Bio-Rad, Hercules, CA, USA). Data were normalized to per mg protein.

### 4.4. Western Blot Analysis

Thirty micrograms of homogenized colon protein lysate (described above) from each sample were separated by SDS-PAGE and electro-transferred to nitrocellulose membrane for immunoblot analyses. The following antibodies were used: anti-S100A9 (73425S, 1:1000; Cell signaling, Danvers, MA, USA), anti-F4/80 (A1256, 1:1000; ABclonal, Woburn, MA, USA), anti-GAPDH (2118S, 1:2000; Cell signaling, Danvers, MA, USA), anti-rabbit IgG, HRP linked secondary antibody (7074, 1:5000; Cell signaling, Danvers, MA, USA). Clarity^TM^ Western ECL Substrate (1705060; Bio-Rad, Hercules, CA, USA) was used for chemiluminescence detection on the ChemiDoc Imaging System (Bio-Rad, Hercules, CA, USA). Protein bands were quantified using ImageJ; the integrated density of protein of interest was normalized to GAPDH, and data were presented as % control.

### 4.5. Cell Culture and Mitochondrial Function Analysis by Seahorse Flux Assay

Human Caco-2 cells were obtained from the American Type Culture Collection (HTB-37; ATCC, Manassas, VA, USA) and were maintained in Dulbecco’s Modified Eagle Medium (DMEM) supplemented with 20% fetal bovine serum (FBS) and 1% penicillin/streptomycin (PS) at 37 °C in an incubator with 5% CO_2_, according to ATCC’s recommended protocol. Mitochondria respiration was assessed using the Seahorse XF Cell Mito Stress Test Kit (Agilent Technologies, North Billerica, MA, USA) according to the manufacturer’s instructions. Caco-2 cells were seeded in an XF 96-well cell culture microplate at 4 × 10^3^ cells/well density in 200 µL of culture medium. After 16 h, cells were treated with saline or 100 nM human AG (031-30; Phoenix Pharmaceuticals, Burlingame, CA, USA) in DMEM with 1% FBS and 1% PS for 24 h. On the day of assay, cells were washed and incubated with 180 μL Seahorse-ADF medium (Seahorse XF Assay Medium, Agilent Technologies, North Billerica, MA, USA) supplemented with 17.5 mM glucose, 2 mM Glutamax, 1 mM sodium pyruvate, and 1% penicillin-streptomycin adjusted to pH 7.4. Extracellular flux analysis using the mitochondrial stress kit was performed to assess mitochondrial respiratory capacity and function as we previously described [26].

### 4.6. PPARγ Reporter Assay

An in vitro reporter assay was used to quantify PPARγ activity, based on the plasmid construct developed by Degrelle et al. [53]: the reporter plasmid carries the PPARγ response element (PPRE) and secreted form of NanoLuciferase sequences (PPRE-pNL1.3[secNluc]; plasmid# 84394, Addgene, Watertown, MA, USA) (Figure 4B; adapted from [53]). Caco-2 cells were seeded and transfected with PPRE-pNL1.3 or pNL1.3 basic secreted luciferase reporter as a control in 24-well plates using Lipofectamine 3000 (Invitrogen, Waltham, MA, USA), following the manufacturer’s protocol. Some 48 h later, the transfected cells were washed and treated with the indicated dose of human AG, or JMV-2959 (Sigma, St. Louis, MO, USA) in DMEM with 1% FBS and 1% PS. 20 μL of supernatant from each well was collected after 2 h of treatment and dispensed into black 96-well plates. The amount of secreted NanoLuc luciferase activity was determined using the Nano-Glo Luciferase Assay (Promega, Madison, WI, USA) according to the manufacturer’s protocol. Luminescence was measured using the microplate reader CLARIOstar Plus (BMG Labtech, Cary, NC, USA).

### 4.7. Statistical Analysis

Results were expressed as mean ± standard error (SE) at a significance level of *p* < 0.05. For the comparison between more than two groups, one-way ANOVA was used if there was a single independent variable (treatment effect) or two-way ANOVA for repeated measures (treatment × time). To further test pairwise differences between treatment groups, Tukey’s post hoc tests were used. Statistical analyses were carried out using GraphPad PRISM 9.2.0 (GraphPad, San Diego, CA, USA).

## Figures and Tables

**Figure 1 ijms-24-00565-f001:**
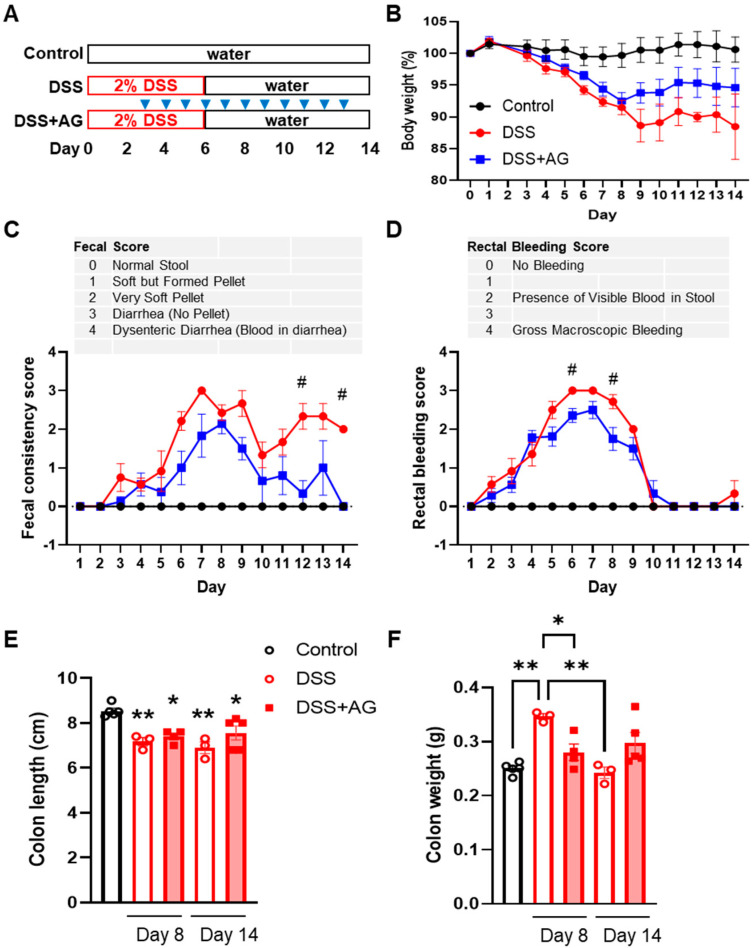
Ghrelin alleviates DSS-induced colitis in 16-18 months-old female mice. Normal or 2% DSS in drinking water was given to mice for 6 days, then switched to normal drinking water until day 14. Mice in DSS + AG group were intraperitoneally injected with ghrelin daily from days 3–13, indicated by blue arrowheads (**A**). Disease activities were monitored, including body weight (**B**), stool consistency (**C**) and rectal bleeding (**D**). Mice were euthanized on day 8 and day 14 after assessing disease activities and their colons dissected, length (**E**) and weight (**F**) measured. For (**B**–**D**), data were analyzed with two-way ANOVA with a mixed model (treatment group × time) followed by Tukey’s post-hoc tests. For (**E**,**F**), data were analyzed with one-way ANOVA (treatment) followed by Tukey’s post-hoc tests. * *p* < 0.05, ** *p* < 0.01, compared to control in E, or as indicated in (**F**); # *p* < 0.05, DSS vs. DSS + AG. AG: acyl-ghrelin; DSS: dextran sulfate sodium.

**Figure 2 ijms-24-00565-f002:**
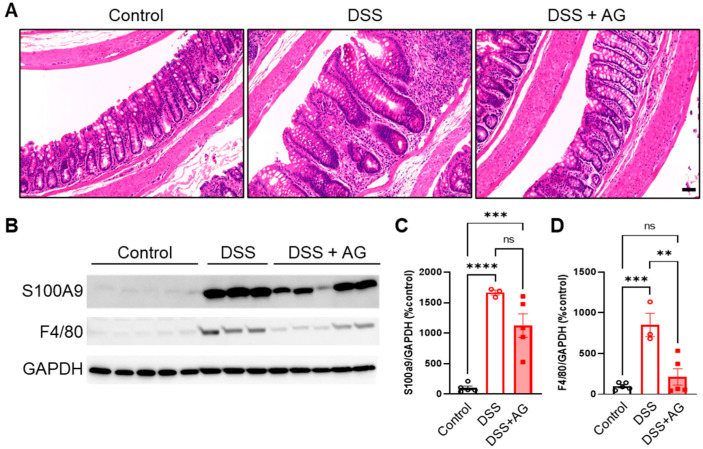
Ghrelin treatment reduces colon inflammation post-induction of DSS-induced colitis in old mice. (**A**) Representative photographs (around the mid-section) of the colons were collected on day 14 from the control, DSS and DSS + AG groups. Scale bar = 50 μm. Colon homogenates were assayed for S100A9 (**B**,**C**) and F4/80 (**B**,**D**) protein levels using western blot analysis. Data were analyzed with one-way ANOVA (treatment) followed by Tukey’s post-hoc tests, ** *p* < 0.01, *** *p* < 0.001, **** *p* < 0.0001. AG: acyl-ghrelin; DSS: dextran sulfate sodium; ns: not significant.

**Figure 3 ijms-24-00565-f003:**
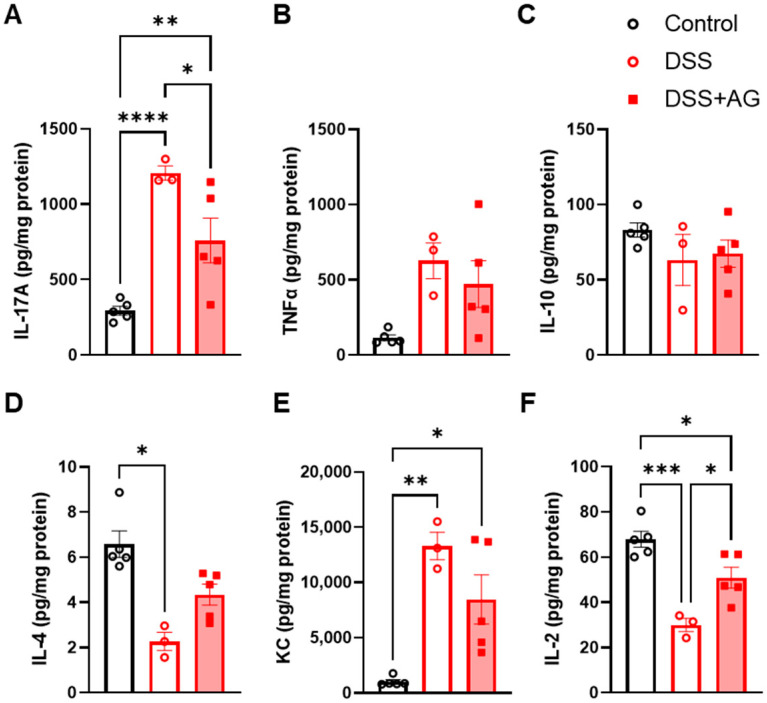
Ghrelin treatment alters inflammatory cytokines in old mice recovering from DSS-induced colitis. Colon samples collected on day 14 were processed and used for cytokine measurements: (**A**) IL-17A, (**B**) TNFα, (**C**) IL-10, (**D**) IL-4, (**E**) KC, (**F**) IL-2. Data were analyzed with one-way ANOVA (treatment) followed by Tukey’s post-hoc tests. * *p* < 0.05, ** *p* < 0.01, *** *p* < 0.001, **** *p* < 0.0001. AG: acyl-ghrelin; DSS: dextran sulfate sodium; KC: keratinocyte chemoattractant; IL: interleukin; TNFα: tumor necrosis factor-alpha.

**Figure 4 ijms-24-00565-f004:**
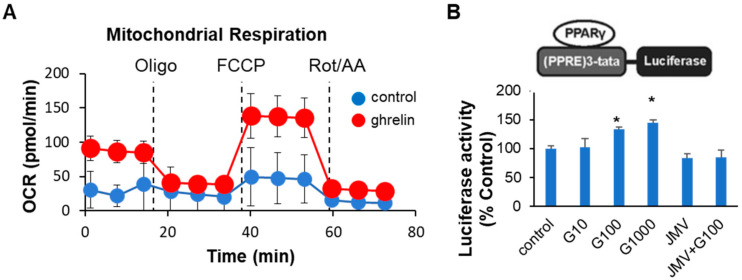
Ghrelin increases mitochondrial respiration and dose-dependently activated PPARγ transcription in Caco-2 cells. (**A**) Caco-2 cells were treated with vehicle or 100 nM ghrelin for 24 h. ATP synthase inhibitor oligomycin (Oligo), mitochondrial uncoupling reagent FCCP, and electron transport chain inhibitors rotenone/actinomycin A (Rot/AA) were added at indicated times. Error bars represent SD in 12 replicates. (**B**) PPRE-luciferase plasmids were transiently transfected into Caco-2 cells for 48 h. Cells were then treated for 2 h with 10, 100 or 1000 nM of human acyl-ghrelin (G). JMV: 3 uM of Growth hormone secretagogue receptor antagonist JMV 2959. Error bars represent SEM, experiments performed in triplicate, repeated 2 times. Data were analyzed with one-way ANOVA (treatment) followed by Tukey’s post-hoc tests. * *p* < 0.05 vs. control group. OCR: oxygen consumption rate; PPRE: peroxisome proliferator response element.

## Data Availability

Data supporting reported results are included in the manuscript.

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
