# Peer review of "Ghrelin Alleviates Experimental Ulcerative Colitis in Old Mice and Modulates Colonocyte Metabolism via PPARγ Pathway"

_ijms, 2022, doi:10.3390/ijms24010565_

Round 1
Reviewer 1 Report
In this submitted manuscript by Muthyala et al studied the protective role of ghrelin in ulcerative colitis. Using dextran sulfate sodium (DSS)-induced mouse model, they showed that ghrelin alleviates rectal bleeding and improves fecal consistency. They further showed that ghrelin treatment reduces colon inflammation at the recovery phase. Taken together, this manuscript reported findings that are meaningful for the field. While the data is solid, this study can be further strengthened by addressing the following minor points:
1. Can the authors validate the potential mechanism shown in the cell model with the mouse model?
2. The authors showed that ghrelin reduces inflammation markers at day14 but not day8, while rectal bleeding of the DSS+AG group showed significance at day6 and day8. What could be the potential mechanism?
Author Response
- Can the authors validate the potential mechanism shown in the cell model with the mouse model?
Response: We agree with the reviewer that it is important to validate the mechanism shown in the cell model with respect to the mouse model. Given the limited availability of aged mice, detailed molecular characterization is beyond the scope of the current manuscript. Further works employing hydroxyprobes to assess epithelial oxygenation levels, isolation of intestinal epithelial cells from mouse models to assess mitochondrial function or gene expression changes in β-oxidation pathways, and characterization of gut microbiota to determine whether ghrelin treatment ameliorates gut dysbiosis will be required to validate our in vitro findings. This statement has been added to the Discussion (lines 250-254).
- The authors showed that ghrelin reduces inflammation markers at day14 but not day 8, while rectal bleeding of the DSS+AG group showed significance at day6 and day8. What could be the potential mechanism?
Response: We thank the reviewer for the insightful comment. While DSS is widely used in mouse model of colitis, the mechanisms by which DSS induces intestinal inflammation and accompanying diarrhea and bleeding are unclear. It has been suggested that negatively-charged DSS forms nano-lipocomplexes with medium-chain-length fatty acids in the colon and induces erosion of the epithelia, ultimately compromising barrier integrity, resulting in increased colonic epithelial permeability and translocation of proinflammatory intestinal contents (bacteria and their products) into underlying tissue [1, 2]. In addition, the anticoagulant property of DSS aggravates intestinal bleeding. Hence, the cessation of DSS from day 6 allows tissue restitution without further aggravation, and this process may precede the clearance of infiltrated immune cells and resolution of inflammation. The observation that ghrelin reduced rectal bleeding on day 6 and day 8 suggests that it may also be involved in this tissue restitution phase, via regulation of proliferation or apoptosis. With respect to our data, Zhang et al. [3] recently reported that ghrelin inhibits intestinal epithelial cell apoptosis in DSS-induced colitis. Interestingly, a recent study demonstrated a novel mechanism by which neutrophil-mediated immunothrombosis and extracellular trap formation promotes resolution of rectal bleeding in ulcerative colitis [4]. Whether ghrelin modulates neutrophil function in terms of immunothrombosis to ameliorate rectal bleeding remains to be determined. This Discussion has been added to the manuscript (lines 200-219).
- Laroui, H.; Ingersoll, S. A.; Liu, H. C.; Baker, M. T.; Ayyadurai, S.; Charania, M. A.; Laroui, F.; Yan, Y.; Sitaraman, S. V.; Merlin, D., Dextran sodium sulfate (DSS) induces colitis in mice by forming nano-lipocomplexes with medium-chain-length fatty acids in the colon. PLoS One 2012, 7, (3), e32084.
- Chassaing, B.; Aitken, J. D.; Malleshappa, M.; Vijay-Kumar, M., Dextran sulfate sodium (DSS)-induced colitis in mice. Curr Protoc Immunol 2014, 104, 15 25 1-15 25 14.
- Zhang, L.; Cheng, J.; Shen, J.; Wang, S.; Guo, C.; Fan, X., Ghrelin Inhibits Intestinal Epithelial Cell Apoptosis Through the Unfolded Protein Response Pathway in Ulcerative Colitis. Front Pharmacol 2021, 12, 661853.
- Leppkes, M.; Lindemann, A.; Gosswein, S.; Paulus, S.; Roth, D.; Hartung, A.; Liebing, E.; Zundler, S.; Gonzalez-Acera, M.; Patankar, J. V.; Mascia, F.; Scheibe, K.; Hoffmann, M.; Uderhardt, S.; Schauer, C.; Foersch, S.; Neufert, C.; Vieth, M.; Schett, G.; Atreya, R.; Kuhl, A. A.; Bleich, A.; Becker, C.; Herrmann, M.; Neurath, M. F., Neutrophils prevent rectal bleeding in ulcerative colitis by peptidyl-arginine deiminase-4-dependent immunothrombosis. Gut 2022, 71, (12), 2414-2429.
Reviewer 2 Report
In this manuscript, author investigated the potential effect of ghrelin on colonocyte metabolism and whether PPARr is a downstream factor mediating the effect of ghrelin. Most of results were interesting to understand a novel role for ghrelin in regulating intestinal epithelial metabolism through PPARr. The data are comprehensive and well presented in the figures, and the experimental approaches appear sound. However, the manuscript needs minor modifications to be considered by International Journal of Molecular Sciences.
Minor comments:
1) Why did you select single dose of AG? You should describe a reason into Result or Discussion.
2) Author should add the limitation of this study and further study in Discussion section.
3) Approval number for animal studies should be inserted into materials and method section.
4) All number should be separated unit except % and oC. Also, unit should be described same pattern.
5) Also, all abbreviation should be fully described when it firstly appeared. Also, this description should be not repeated in text.
6) The information for all products, reagents and machines used this study should be clearly described based on the guide of journal. Ex) Product name (Company, Region, Country).
7) Figure legends should be corrected to include title, information for figure, and information for abbreviation. Title should be changed from sentence to phrase.
8) References should be corrected according to journal guideline.
Author Response
1) Why did you select single dose of AG? You should describe a reason into Result or Discussion.
Response: The reason for choosing a dose of 500 μg/kg acyl-ghrelin in this study was provided in the Introduction (lines 58-60): “We previously showed that 500 μg/kg acyl-ghrelin treatment alleviated fasting-induced muscle atrophy in old male mice [26]. Here, we used the same dose of acyl-ghrelin to test its efficacy in attenuating DSS-induced colitis in old female mice.”
2) Author should add the limitation of this study and further study in Discussion section.
Response: We acknowledge that using a single dose of AG to test its efficacy in old mice is a limitation of this study. Future studies employing a more comprehensive dosage range will be carried out to assess ghrelin’s therapeutic effects and validate the potential mechanisms in vivo. This has been added to the Discussion section (lines 275-278). Other directions for further study has been mentioned throughout the Discussion section (lines 218-219; lines 247-254).
3) Approval number for animal studies should be inserted into materials and method section.
Response: We have inserted the approval number as suggested.
4) All number should be separated unit except % and oC. Also, unit should be described same pattern.
Response: We have revised the text and corrected the identified errors throughout the text.
5) Also, all abbreviation should be fully described when it firstly appeared. Also, this description should be not repeated in text.
Response: We have revised the text and corrected the identified errors throughout the text.
6) The information for all products, reagents and machines used this study should be clearly described based on the guide of journal. Ex) Product name (Company, Region, Country).
Response: This information has been updated. For key reagents, the catalogue number has also been included in the following format: Catalogue number, Company, Region, Country.
7) Figure legends should be corrected to include title, information for figure, and information for abbreviation. Title should be changed from sentence to phrase.
Response: We have revised the figure legends to include information for abbreviations as suggested. For titles, we prefer to leave them as they are; this seems to be acceptable as other research articles in the special issue in Intl J Molecular Sciences also adopt this style of figure legends (for example, see https://www.mdpi.com/1422-0067/23/5/2667).
8) References should be corrected according to journal guideline.
Response: The references have been prepared with the bibliography software package EndNote, using the Style file for Intl J Molecular Sciences according to journal guideline.